# Magnetofermionic condensate in two dimensions

L.V. Kulik[1], A.S. Zhuravlev[1], S. Dickmann[1,2], A.V. Gorbunov[1], V.B. Timofeev[1], I.V. Kukushkin[1] & S. Schmult[3]

Coherent condensate states of particles obeying either Bose or Fermi statistics are in the focus of interest in modern physics. Here we report on condensation of collective excitations with Bose statistics, cyclotron magnetoexcitons, in a high-mobility two-dimensional electron system in a magnetic field. At low temperatures, the dense non-equilibrium ensemble of long-lived triplet magnetoexcitons exhibits both a drastic reduction in the viscosity and a steep enhancement in the response to the external electromagnetic field. The observed effects are related to formation of a super-absorbing state interacting coherently with the electromagnetic field. Simultaneously, the electrons below the Fermi level form a super-emitting state. The effects are explicable from the viewpoint of a coherent condensate phase in a non-equilibrium system of two-dimensional fermions with a fully quantized energy spectrum. The condensation occurs in the space of vectors of magnetic translations, a property providing a completely new landscape for future physical investigations.

[1] Laboratory of Non-equilibrium Electron Processes, Institute of Solid State Physics, RAS, 142432 Chernogolovka, Russia. [2] Federal University of Rio Grande do Norte, International Institute of Physics, Anel Viário da UFRN, s/n—Lagoa Nova, Natal-RN 59078-970, Brazil. [3] Department of Low Dimensional Electron Systems, Max-Planck-Institut für Festkörperforschung, Heisenbergstrasse 1, 70569 Stuttgart, Germany. Correspondence and requests for materials should be addressed to L.V.K. (email: kulik@issp.ac.ru).

Nowadays the search for new physical systems exhibiting macroscopic quantum coherency remains among the mainstream research activities. In the quantum condensate state huge numbers of particles behave coherently with their motion described by a single macroscopic wave function. The practical realization of the condensate states yields fascinating effects, such as superfluidity in liquid $^4$He and $^3$He (ref. 1) or superconductivity in various materials[2]. Despite the ongoing search for new systems, reliably established cases of a transition into the coherent condensed state are fairly scarce.

Transitions into a coherent condensed state are divided into two fundamental groups, with one of these described by the phase transition into the thermodynamic equilibrium state, such as Bose–Einstein condensation in superfluid $^4$He. Another group that has recently attracted particular attention is represented by so-called non-stationary condensates, that is, systems driven from equilibrium by an external force. Despite the fact that no detailed equilibrium is attained, the system can be split into macroscopic subsystems in which a local quasi-equilibrium is achieved, with the subsystems remaining in the quasi-equilibrium state for a sufficiently long time required for occurrence of condensation. Exciton-polariton condensates[3,4], magnon condensates[5], atomic condensates[6,7] and dipolar exciton condensates[8–10] can be classified as non-equilibrium.

A special case among condensate states is fermion condensates that can also be divided into thermodynamically equilibrium cases, such as superconductors, $^3$He, and a state with total filling factor $v = 1$ in the double electron layers[11,12], and non-equilibrium cases, such as condensates of spin-polarized cold $^{40}$K (ref. 13) and $^6$Li (ref. 14) atoms with Fermi statistics.

Particular interest is attracted to the condensation in a two-dimensional (2D) fermion system driven from equilibrium by emergence of an ensemble of collective excitations with Bose statistics (magnetoexcitons)[15–18]. Owing to the specific features of the 2D electron energy spectrum, the quantizing magnetic field turns a 2D metal into a quantum Hall insulator, provided that the electrons fill an even integer number of Landau spin sublevels. Excitations in quantum Hall insulators are cyclotron magnetoexcitons (CMEs) obeying Bose statistics; these are composed of an electron excited into the unfilled Landau level and a Fermi-hole (an electron vacancy) in the conduction band left below the electron Fermi level[19] and are called by analogy with excitons in semiconductors consisting of an electron in the conduction band and a hole in the valence band. The presence of a boson component in the correlated fermion system makes CMEs potential candidates for macroscopic non-equilibrium condensates.

One of the simplest realizations of CME in a 2D electron system in a high magnetic field is achieved on photoexcitation of an electron from the zero to the first Landau level in the integer quantum Hall state with two completely filled spin sublevels of the zero Landau level. There are two types of CMEs: a spin singlet, $S = 0$, where $S$ is the spin quantum number of the magnetoexciton, and a spin triplet, $S = 1$, with spin projections along the magnetic field axis $S_z = -1, 0, 1$. The optically active spin singlet CME is a magnetoplasmon[20]. It is annihilated upon excitation, rapidly emitting an electromagnetic field quantum with cyclotron energy $\hbar\omega_c$ (ref. 21). Spin-triplet CMEs are not optically active. They are 'dark' magnetoexcitons with relaxation times reaching 100 μs and higher[22–24], which is a record value for quasi-particles in a 2D quantum system formed from free electrons. The aforementioned property allows creating a dense ensemble of such quasi-particles by optical spectroscopy means.

Spin-triplet CMEs are bosons, so a dense ensemble of these is likely to exhibit the phenomenon of Bose-condensation. However, it is known that in 2D and 1D systems, the long-range order is destroyed by density fluctuations at any finite temperature[25]. For this reason, Bose–Einstein condensates can exist at zero temperature only. To be more precise, the long-range order in the 2D case may not be destroyed completely owing to the interparticle interactions. Power law spatial correlations are retained and lead to a Berezinsky–Kosterlitz–Thouless (BKT) phase transition at finite temperatures[26,27]. There is now experimental evidence of the BKT transition in quasi-2D systems, such as layered superconductors, liquid helium films, and Josephson junction arrays. For excitonic systems BKT phase transition was considered in the theoretical papers[15–18].

In the following, we present experimental evidence for bosonization of the 2D fermion system and finite-temperature condensation in a pure 2D electron quantum liquid. Within the dense ensemble of non-equilibrium long-lived spin-triplet CMEs a phase transition into a coherent condensed state occurs at temperatures below 1 K. The CME phase exhibits a drastic reduction in viscosity and a steep enhancement in the response to the external electromagnetic field.

## Results

**Monitoring the magnetoexciton ensemble**. We employ the resonant photoexcitation technique to create a macroscopic non-equilibrium ensemble of spin-triplet CMEs. However, this is not sufficient. It is also necessary to develop a method of monitoring the properties of the ensemble when varying the system parameters, such as electron temperature $T$, photoexcitation power density $P$, and magnetic field $B$. The main difficulty is that spin-triplet CMEs are 'dark' quasi-particles, that is, they do not interact with the external electromagnetic field in the dipole approximation. To monitor the state of the non-equilibrium CME ensemble, we developed an experimental technique of photoinduced resonant reflection (PRR), which utilizes an electromagnetic field in a frequency range that greatly exceeds the CME energy (Fig. 1a,b). Optically allowed spatially direct transitions from the heavy-hole valence band to the conduction band enable to test separate parts of the CME: the excited electron above and the Fermi-hole below the Fermi level. Although triplet CMEs themselves are not optically active, direct transitions from the valence band to the excited electron states and Fermi-hole states are allowed. The study of the PRR signals of the Fermi-holes and/or excited electrons enables us to draw conclusions about the behaviour of the entire CME ensemble[24].

**Phase transition**. In the temperature range close to 1 K, we observe a phase transformation of the non-equilibrium electron system with a number of unusual properties. The diffusion rate for the CMEs demonstrates a non-monotonic temperature dependence. When the temperature decreases below 1.5 K, the relaxation time $\tau$ first increases exponentially, indicating the collection of the non-equilibrium CMEs in the excitation spot. The relaxation time then drops suddenly (Fig. 1c). The effect points to a threshold reduction in the viscosity of the CME ensemble upon diffusion of the CMEs out of the excitation spot. The CMEs spread out through the sample. With further reduction of temperature, the relaxation time returns to its initial value (Fig. 1c; Supplementary Fig. 1).

Since the transport properties around the phase boundary change markedly, and the number of CMEs in the excitation spot varies accordingly, it is reasonable to find an intrinsic property of the non-equilibrium ensemble related to the order parameter of the phase transition that changes continuously and monotonically at the phase transition boundary. Such a property appears to be the susceptibility of a single Fermi-hole included into CME to the external electromagnetic field. It should be noted

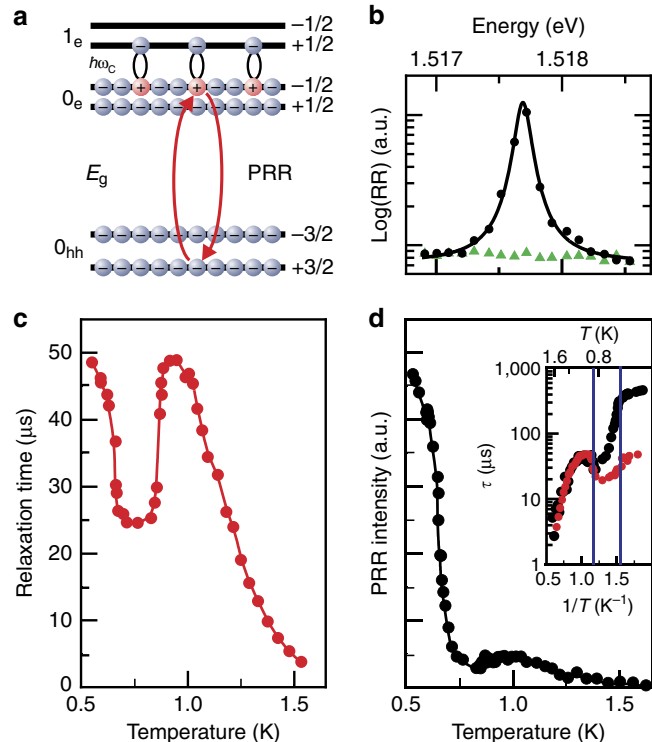

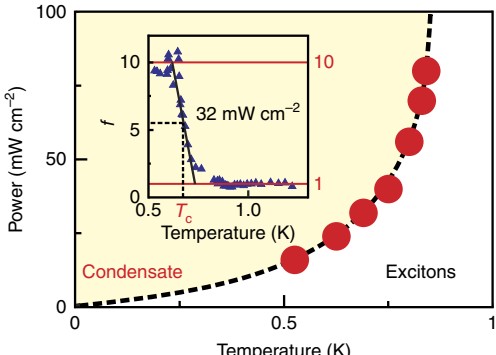

**Figure 2 | Phase diagram of CME condensation.** Phase boundary separating incoherent gas and condensate of spin-triplet CMEs (dots). The saturation of the phase boundary at large photoexcitations is due to a limited amount of CMEs that the electron system can accumulate (around 0.15 of Landau spin sublevel degeneracy number $N_\Phi$). The dashed line serves solely as an eye cursor. The inset illustrates how the critical temperature is chosen (half the height of the threshold increase of oscillator strength $f(T)$).

**Figure 1 | Phase transition in non-equilibrium 2D electron system.** (**a**) Diagram of 2D single-electron energy states in magnetic field ($\pm 1/2$, and $\pm 3/2$ are the spins of the electrons in the conduction band and heavy holes in the valence band). The black ellipses illustrate schematically how the photoexcited Fermi-holes in the zero electron Landau level ($0_e$) couple with the photoexcited $1_e$ electrons to form spin-triplet CMEs. The red arrows denote the optical transition ($0_h - 0_e$) of photoinduced resonant reflection (PRR). (**b**) Resonant reflection (RR) spectrum of the equilibrium 2D electron system (green triangles) and that of the system driven from equilibrium by photoexcitation of the CME ensemble (black dots). The solid line is a Lorenz approximation to the experimental points. (**c**) Temperature dependence of CME relaxation time $\tau(T)$ (dots). The solid line is drawn for convenience. (**d**) Temperature dependences of PRR intensity $I(T)$ in the conditions of stationary photoexcitation (dots). The solid line is drawn for convenience. The inset presents comparison between $\tau(T)$ (red dots) and $I(T)$ (black dots) in the logarithmic scale ($I(T)$ is multiplied by a constant so that $\tau(T)$ and $I(T)$ coincide at 1 K).

that the CME density under the quasi equilibrium conditions is directly proportional to relaxation time $n(T) \propto \tau(T)$. Moreover, the PRR intensity $I(T)$ is directly proportional to $n(T)$: $I(T) \propto f(T)n(T)$, where $f(T)$ is the oscillator strength of the corresponding optical transition. Collecting the data for these two measurable quantities, $\tau(T)$ and $I(T)$ (Fig. 1c,d), enables us to find the susceptibility of a single Fermi-hole to the external electromagnetic field $f(T) \propto I(T)/\tau(T)$. This quantity does not depend on the number of CMEs and behaves monotonically in the entire range of electron temperatures (Fig. 2). It is almost independent of temperature down to the phase transition temperature, as expected for an ensemble of incoherent CMEs (Fig. 2; Supplementary Note 1). However, with further reduction of temperature, a gigantic rise in the oscillator strength is observed. The Fermi-holes start oscillating coherently under electromagnetic excitation (super-absorption), which can be described in terms of CME condensation.

**Condensed phase properties.** Resonant reflection involves two sequential processes: the resonant absorption of a photon and the

emission of the same photon returning the electron system to the initial state. It is therefore sufficient to consider the first process of resonant absorption. The single CME is an eigenstate of the system under study (see the details in the Supplementary Note 1). This state is described by the action of the exciton creation operator $\mathcal{Q}_\mathbf{q}^\dagger$ on the ground state $|0\rangle$ and is characterized by the vector of magnetic translations (pseudomomentum) $\mathbf{q}$ (refs 28,29) and energy $\hbar\omega_c + g\mu_B B S_z + \mathcal{E}_\mathbf{q}$, where $g$ is the $g$-factor, $\mu_B$ the Bohr magneton, and $\mathcal{E}_\mathbf{q}$ the Coulomb contribution[19]. However, the entire CME ensemble is not an eigenstate owing to the presence of inter-excitonic interactions. For example, such interactions, even if weak, play a decisive role in the formation of the BKT condensate. In the present communication, we do not claim a theoretical description of the observed phase transition, as there are at the moment no appropriate mathematical models describing the inter-excitonic interaction, but study only the process of resonance absorption within the framework of a simplified model of non-interacting CMEs in two opposite cases: far above and far below the phase transition temperature.

At comparatively high temperatures or low CME densities $n$, the non-equilibrium state is described by an incoherent ensemble of $N$ CMEs with different pseudomomenta $\mathbf{q}_i$ ($i = 1, 2, ..., N$, where $N = 2\pi l_B^2 n N_\Phi$, $l_B = \sqrt{\hbar c/eB}$ is the magnetic length, $N_\Phi$ is the Landau spin sublevel degeneracy number). The resonant absorption by the $N$-CMEs state is reduced to a single-CME process, where the CME is transformed into an exciton formed by the excited electron and heavy valence hole (Fig. 1a). The square of the matrix element for this optical transition is temperature-independent and is approximately equal to $1/N_\Phi$. Consider now an extremely coherent case, where $N$ CMEs are in a single-quantum state described by condensate $\left(\mathcal{Q}_\mathbf{q}^\dagger\right)^N |0\rangle$. The square of the transition matrix element for the resonant absorption is equal to $N/N_\Phi$. Applying these considerations to the CME states confined in the real 2D random potential, we relate the observed growth of the resonant reflection intensity to the macroscopic CME occupation of a single $q_i$ state (see details in the Supplementary Note 1).

**Reconstruction of electron vacuum.** It is obvious that the Fermi-holes involved in the spin-triplet CMEs cannot be coherent

as they are just empty states in the electron Fermi sea. Rather, the electrons separating the Fermi-holes are rearranged in such a way that the Fermi-hole ensemble behaves coherently on interaction with the external electromagnetic field. This observation suggests that the estimate made in the Supplementary Note 1 is a reasonable approximation only. The theory would be exact provided that the electron system below the Fermi level (electron vacuum) remained unperturbed on excitation of the non-equilibrium ensemble. Yet, the optical response of the electron vacuum itself reveals that it is strongly modified on creation of CMEs.

The optical response of the electrons below the Fermi level is verified by the photoinduced emission (PE) technique, that is, photoluminescence in the presence of a non-equilibrium CME ensemble (Figs 3 and 4). In contrast to the PRR with its single optical transition, PE allows two possible recombination channels (Fig. 3). The first is related to the recombination of an electron from the conduction band with a heavy hole located far away from the CME. This recombination channel is responsible for the $\sigma^+ - \sigma^-$ doublet with a spectral separation of lines equal to the sum of spin-splittings for the electrons in the conduction band and heavy holes in the valence band observed in the luminescence spectra (Fig. 3). The second channel is related to the recombination of a hole located in close proximity to the CME. The first

recombination channel dominates at elevated temperatures above 1 K, whereas the other, non-equilibrium channel starts dominating at the temperature below that of the phase transition (Fig. 3). Although the maximum fraction of the spin-triplet CMEs does not exceed $0.15N_\Phi$ (ref. 24), the non-equilibrium recombination channel finally picks up nearly the total oscillator strength of all optical transitions (super-emission; Figs 3 and 4). It should be emphasized that the spectral position of this luminescence channel does not coincide with any optical transition in the unperturbed single-particle pattern.

## Discussion

The results obtained imply that here a more complex physical phenomenon than condensation of Bose particles is present. This phenomenon can be described as a coupling of photoexcited spin-up and spin-down electrons at the Fermi surface, which reconstructs the electron system into a new state, magnetofermionic 2D condensate (a condensate in the fermion 2D system with a spectrum totally quantized by the magnetic field). The magnetofermionic condensate interacts coherently as a whole with the external electromagnetic field and appears to be both a super-emitting and super-absorbing medium (see the scheme in Fig. 4).

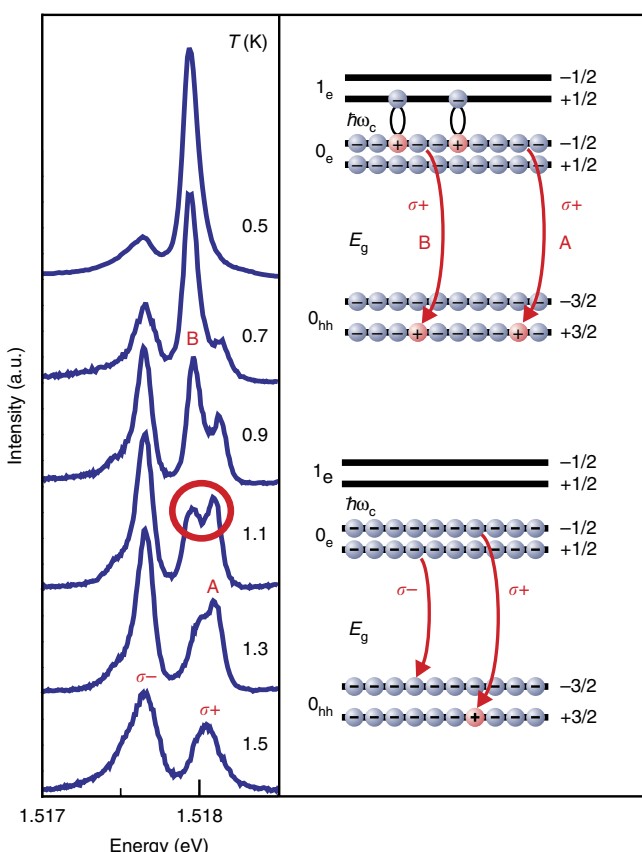

**Figure 3 | Photoinduced emission—temperature dependence.** Emission spectra corresponding to radiative recombination of electrons below the Fermi level with heavy holes measured in the temperature range $T = 0.5$–$1.5$ K at fixed photoexcitation power $P = 32$ mW cm$^{-2}$. At elevated temperatures a single optical transition in $\sigma^+$ polarization (A) is seen. The B line observed at low temperatures is a CME signature. The bottom diagram demonstrates optical transitions between the single-particle states in two possible polarizations of the emitted photons. The top diagram shows expected optical transitions in $\sigma^+$ polarization in the presence of a CME ensemble.

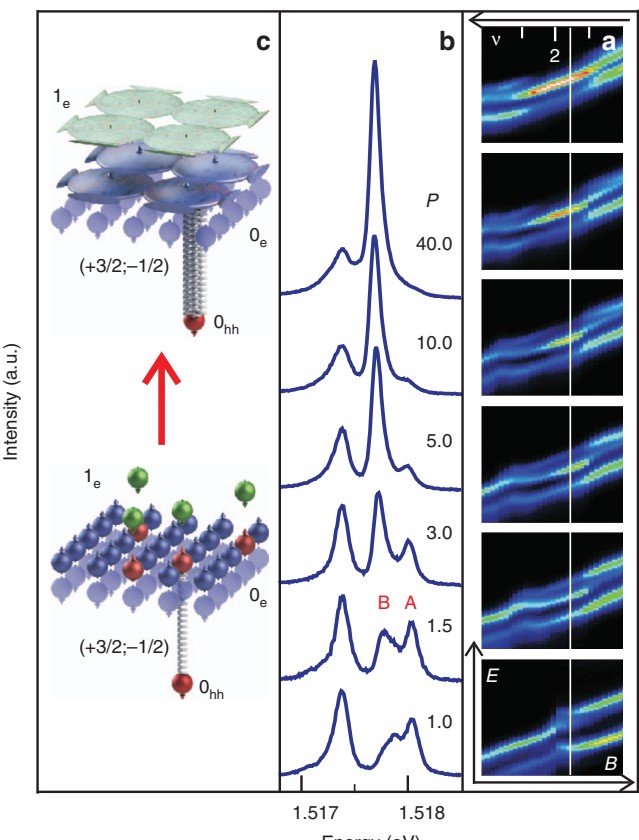

**Figure 4 | Photoinduced emission—pumping power dependence.**
(**a**) Evolution of photoinduced emission spectra in magnetic field. No condensate occurs beyond the range of $\nu = 2 \pm 0.15$ indicated by the white ticks. The straight white line denotes the image cross-section shown in **b** (photoinduced emission spectra at fixed temperature of 0.5 K and different photoexcitation powers $P$ (in mW cm$^{-2}$)). The maxima of the optical transition in $\sigma^+$ polarization are shifted to a single-energy value.
(**c**) Modification of the non-equilibrium electron system during formation of magnetofermionic condensate.

The magnetofermionic condensate exhibits few similarities with and shows some differences from the known coherent Bose-condensed states. First of all, condensation occurs neither in the reciprocal space where celebrated Bose–Einstein condensates form[30], nor in the real space where famous exciton liquid (electron-hole droplets) exists[31]; it occurs in the space of vectors of magnetic translations. There are obvious similarities with the condensate of excitons predicted in refs 15,16. However, the observed reconstruction of the Fermi surface limits this similarity. This reconstruction suggests an analogy to superconductors. Yet, the existence of a rigid cyclotron gap, the necessity for a strong magnetic field, the charge neutrality of CMEs and the type of coupling (in superconductors, two electrons with opposite spins are bound in a Cooper pair by the electron–phonon interaction, whereas in the magnetofermionic condensate, one spin-up electron couples with many spin-down electrons via the Coulomb interaction) makes this analogy subtle. In addition, the arguments similar to those for the BKT transition need to be found to explain why condensation occurs at the finite temperature. Hopefully, further experimental and theoretical studies will enrich the current understanding of this pure 2D-fermion state with bosonic properties.

## Methods

**Samples.** We investigated two high-quality heterostructures (the dark mobility was $15 \times 10^6 \, \text{cm}^2 \text{V}^{-1} \text{s}^{-1}$) containing a symmetrically doped GaAs/AlGaAs single quantum well with a width of 35 nm and nearly equal electron concentrations in the 2D channel of $2 \times 10^{11} \, \text{cm}^{-2}$, but with CME relaxation times differing by one order of magnitude. Qualitatively, the experimental results are similar for both samples, but the pumping power values required to reach condensate states are different. The results for sample #1 with a shorter CME life time are presented in the main body of the manuscript. The results for sample #2 are shown in Supplementary Figs 1 and 2.

**Experimental set-up.** A $3 \times 3 \, \text{mm}^2$ sample was placed into a pumped cryostat with liquid $^3$He, which in turn was placed into a $^4$He cryostat with a super-conducting solenoid. The set-up allowed measurements at bath temperature down to 0.45 K and at magnetic field up to 14 T. Two continuous wave tunable lasers with narrow spectral widths of emission lines (20 and 5 MHz) were employed as optical sources, enabling us to use one of the lasers for resonant excitation of the electron system and the other for recording the resonance reflectance and photo-luminescence spectra. The optical studies were performed using multimode quartz glass fibres with a core diameter of 400 μm and numerical aperture of 0.39.

**Spectral measurements.** One of fibres was used to excite the electron system below the Fermi level by a probing laser to detect the Fermi-holes in the electron density (Supplementary Fig. 2). The pumping of the CME ensemble was carried out by another laser using the same fibre. The total laser power was limited to 0.3 mW to minimize heating effects. The probing laser power was two orders of magnitude smaller than the pumping power. Therefore, the probing laser was considered to be non-disturbing for the non-equilibrium electron system. The second fibre was used for collection of the reflected light from the sample and its transfer onto the entrance slit of a grating spectrometer equipped with a cooled CCD camera. The reflected beam axis coincided with the receiving fibre axis at the incidence angle of 10°. The contribution of the reflection from the sample surface was suppressed using crossed linear polarizers set between the sample and the ends of the probing and collecting fibres (Supplementary Fig. 2). The resonant reflectance spectrum was obtained by scanning the emission wavelength of the probing laser and recording the reflected laser line intensity. The PRR spectrum was obtained as the difference in the resonant reflectance spectra acquired with the pump on and off.

**Kinetic measurements.** For kinetic studies, the pumping laser emission was modulated periodically using a mechanical chopper (a rotating disk with a radial slit). The modulation period was $\sim$11 ms, and the pump pulse rise/decay time was less than 2 μs. The laser beam was focused onto the chopper disk surface by a microscope objective to shorten the pump pulse edge. The wavelength of the probing laser was set to the maximum of the PRR spectrum (Fig. 1b). The probing laser emission reflected from the sample was passed through a narrow-band interference filtre with a spectral width of 1.1 nm to cutoff the pumping laser light and then focused on an avalanche photodiode operating in the photon counting regime. A gated photon counter was used to accumulate the PRR signal as a function of the time delay from the pump cutoff event to obtain a PRR decay curve[24].

**Effect of pumping power on relaxation rate.** The results of the kinetic measurements made in the course of a monotonic temperature change enabled us to obtain the temperature dependences of the relaxation time $\tau(T)$ at different pumping powers (Supplementary Fig. 1). In all the cases a non-monotonic behaviour of $\tau(T)$ was observed. We note that the relaxation time does not separate the relaxation of a CME to the ground state and the diffusion of the same CME out of the excitation spot. However, since the relaxation to the ground state is governed by a much larger energy scale, the only way to reduce the relaxation time for the CMEs by varying temperatures in such a narrow range as that observed in the experiment, is to move out of the excitation spot. The diffusion enhancement is predictively sensitive to the pumping power (that is, to the CME density). It occurs at lower temperatures when the CME density is reduced (Supplementary Fig. 1).

**PRR with spatial separation.** The hypothesis of a higher diffusion rate for condensed CMEs is verified directly by the experiment with spatial separation. A three-fibre technique was used (Supplementary Fig. 2). One extra fibre supplied laser light to pump the CME ensemble just under the fibre, whereas two other fibres were spatially separated from the excitation spot by a distance of 2 mm and were used to measure the resonance reflection from the CME Fermi-holes. Once the CME ensemble was formed, it was possible that some dark CMEs would diffuse over the 2 mm spacing and be observed in the resonance reflection. However, at elevated temperatures such diffusing events were not detected owing to the background parasitic reflection from the sample surface. Below the phase transition temperature, a drastic enhancement of the reflected signal was observed, indicating that a large number of CMEs flowed from the excitation spot to the location of the resonant reflection detection set-up (see the graph in Supplementary Fig. 2). The threshold change in the diffusion is explicable in terms of condensation in the pseudomomentum space. The dark CMEs are $p$-type excitons with a roton minimum in the dispersion curve[19]. When CMEs are in the incoherent state, the absolute values of their dipole moments are nearly equal (close to that of the roton minimum), but the orientations are randomly distributed. Each individual CME easily finds a stop place in the surrounding random potential and stays there until the relaxation to the ground state (see the theory in the Supplementary Note 1). As the pseudomomentum is proportional to the dipole momentum of the CME, the condensation in the pseudomomentum space is simply the formation of a macroscopic unidirectional dipole momentum for the entire system (Supplementary Fig. 3).

**Data availability.** The data presented in this study are available from the corresponding author on request.

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

## Acknowledgements

We are grateful to Professor E.I. Rashba for reading of the manuscript and very useful comments. We also acknowledge the assistance of V.A. Kuznetsov in performing the experiments. The work was supported by the Russian Science Foundation: grant #16-12-10075.

## Author contributions

L.V.K., A.V.G., A.S.Z., V.B.T. and I.V.K. planned the experiments. L.V.K., A.V.G. and A.S.Z. performed the experiments and analysed the data. S.S. prepared the samples. L.V.K., A.V.G. and V.B.T. wrote the main manuscript text. S.D. wrote the theoretical part of the manuscript. L.V.K. and A.S.Z. prepared the figures. All authors have approved the final version of the manuscript.

## Additional information

**Competing financial interests:** The authors declare no competing financial interests.

**Publisher's note**: 

