## [Peer Review File · Nature Communications]

Reviewer #1 (Remarks to the Author):

The manuscript "Magnetofermionic condensate in two dimensions" by L.V. Kulik, A.S. Zhuravlev, S. Dickmann, A.V. Gorbunov, V.B. Timofeev, I. V. Kukushkin, and S. Schmult is devoted to the studies of the condensation of collective excitations with 16 Bose statistics, magnetoexcitons, in a high-mobility two-dimensional electron system in a magnetic field. It was shown that at low temperatures, the nonequilibrium ensemble of magnetoexcitons exhibits both a drastic reduction in the viscosity and a steep enhancement in the response to the external electromagnetic field. According to the manuscript, the observed effects are related to the formation of a superabsorbing state interacting coherently with the electromagnetic field. Besides, the electrons below the Fermi level form a superemitting state. The authors claimed that the observed effects are caused by formation of a coherent condensate phase in a non-equilibrium system of two-dimensional 2D fermions with a fully quantized energy spectrum. The authors concluded that the condensation occurs in the space of vectors of magnetic translations.

While the topic of this manuscript is very interesting, there are several points which have to be clarified.

I suggest for the authors to demonstrate broader familiarity with the literature devoted to this topic.

1. For example, the following papers on dipolar magnetoexcitonic condensates in coupled quantum wells have not been cited:

a. D. Yoshioka and A. H. MacDonald, *J. Phys. Soc. Jpn.* 59, 4211 (1990).

b. Yu. E. Lozovik, O. L. Berman, and V. G. Tsvetus, *Phys. Rev. B* 59, 5627 (1999).

2. The theory of condensation of dipolar magnetoexcitons in a graphene bilayers was reported in O. L. Berman, Yu. E. Lozovik and G. Gumbs, *Phys. Rev. B* 77, 155433 (2008).

3. The theory of condensation of microcavity magnetopolaritons formed by magnetoexcitons in either a semiconductor quantum well or a graphene monolayer, coupled to microcavity photons was presented in O. L. Berman, R. Ya. Kezerashvili, and Yu. E. Lozovik, *Phys. Rev. B* 80, 115302 (2009).

The authors should address how their results on condensation of magnetoexcitons are related to and different from the results, reported in the papers mentioned above.

In summary, the topic and questions touched in the manuscript are new and timely. However, the reported results of the studies have not been compared in details with the earlier papers on this subject (for example, papers mentioned above). I do not recommend this manuscript for publication in *Nature Communications* in its present form until the questions and remarks listed above will be clarified and addressed.

Reviewer #2 (Remarks to the Author):

This paper presents experimental evidence of a new type of state possibly consisting of Bose condensed magnetoexciton excitations of the two-dimensional electron gas of the quantum Hall insulator close to filling factor 2, where in the ground state the lowest Landau Level (LL0) conduction electron states with both spins are filled. Magnetoexciton condensation will occur in the space of magnetic translation vectors and thus differs from real or k-space condensation in other systems. This magnetoexciton state is created away from equilibrium and is accompanied by a change in the properties of the rest of the electrons in the system. Magnetoexcitons in a spin-triplet state are not optically active and are long-lived, which is favorable for condensation in a non-equilibrium state. The authors use resonant photoexcitation to drive spin-triplet magnetoexcitons that can condense and then use photo-induced resonant reflection at frequencies close to the gap energy of conduction electron and valence heavy holes to measure the effects of the possible condensation of the inter-LL conduction electron magnetoexcitons, which are dark and thus not directly observable with light. The probe signal comes from optical transitions between the valence and conduction bands. In particular, a valence electron can be promoted in the LL0 states that are emptied due to the magnetoexciton formation. The authors observe a Lorentzian peak in the photoinduced interband resonant reflection when the non-equilibrium magnetoexciton state is excited. This peak is absent in the equilibrium system. The temperature dependence of this Lorentzian peak demonstrates non-monotonic behavior below 1.5K, which the authors attribute to the formation of a new phase. They also measure photoinduced emission in the presence of non-equilibrium magnetoexcitons, which shows a new peak at low temperatures that the authors attribute to the rearrangement of the conduction electrons already present in the ground state. This is in addition to the possible magnetoexciton condensation and thus the authors call the new state a magnetofermionic condensate. I think that there is enough evidence of a non-equilibrium state, although it is not exactly clear that this state is indeed a macroscopic condensate. I thus recommend publication in Nature Communications. While the authors use a model of noninteracting magnetoexcitons, I note that interactions between driven magnetoexcitons and interband excitons have been observed and studied before in the literature in the nonlinear four-wave-mixing signal during femtosecond timescales. Such time-resolved coherent spectroscopy experiments and the picture of nonlinear interactions between the magnetoexciton and exciton excitations of interest here might be useful to clarify the nature of this new state, since they are observed down to temperatures of Kelvin in the quantum Hall insulator.

Reviewer #3 (Remarks to the Author):

This work investigates the Bose-Einstein condensation of two-dimensional excitons in a strong magnetic field formed of the electron photoexcited to the first Landau level with spin $-1/2$ and a vacancy with spin $+1/2$ in the conduction zone, which together build-up an exciton in the triplet state. Overall, this is potentially a good paper describing very interesting experimental evidence of the transition to a coherent condensed state and presenting a new experimental technique. The work constitutes a good scientific routine, with a significant essence of novelty. In fact, it looks as the authors pretend to originate a novel direction within the scope of coherent condensate states of particles obeying either Bose or Fermi statistics. In this light, the present paper has a solid potential to be interesting to other researchers in the field and for a broad audience of the physics community. For this reason, I recommend it for publication, however, I do have some comments/suggestions that are necessary to making the manuscript more accessible and to improve the presentation, as summarized below.

Comments

1) While the authors pretended to provide a comprehensive overview in the Introduction section, I do not think that the current overview is satisfactory and the authors are unfair in their treatment of previous literature. In terms of the Bose-Einstein condensation of two-dimensional magnetoexcitons a bunch of novel experimental and theoretical papers appeared within the last decade is missed. There have been a comprehensive theoretical studies of the Bose-Einstein condensation (BEC) of two-dimensional magnetoexcitons, which in particular discussed the Berezinsky-Kosterlitz-Thouless phase transition at finite temperatures, and BEC of 2D magnetoexcitons for the conditions of GaAS, bilayer exciton condensates and heterostructures, published by Moskalenko, Liberman et al. [1-5]. The authors use the same name "magnetoexciton" for the photoexcited electron/vacancy complex therefore they must clearly explain the difference between their approach and the theory of BEC of 2D magnetoexcitons developed in [1-5]. References [1-5] are necessary, otherwise the text as it is presented is confusing for readers.

2) The first part of Introduction repeats (in fact it copies) what is written in the first half of the Abstract. I would suggest to remove lines 10-15 in the Abstract. In line 23 should be "...system of two-dimensional (2D) fermions"..

3) There are references [19], [23], [24] in the list of references, but there is no these references in the text. Ref. [19] is probably missed on lines 62/63 and so on.

4) Ref. [13] is probably missed on line 46, as well references on the publication by Moskalenko et al. are needed on line 44 concerning "Excitations in quantum Hall insulators are magnetoexcitons..."

5) All the same I believe that there is unnecessary citation [14] - (lines 51-53): "There are two types of magnetoexcitons: a spin singlet, $S = 0$, and a spin triplet, $S = 1$, with spin projections along the magnetic field axis $S_z = -1, 0, 1$ ", which is a simple statement, which does not require reference [14].

6) Check Ref.[8] - line 157, should be [9] which is missed and [6, 7] in the list of references below.

7) Line 134 the authors wrote: "This observation suggests that the theory developed in the previous paragraph is a reasonable approximation for the properties of the non-equilibrium system in question". I would suggest to change ".. the theory developed in the previous paragraph.. "for"..the estimate above..."

Overall, an appropriate literature overview is needed to clarify the pose of this particular submission among other works. This will not deteriorate the publication chance, but - in contrast - will underline and justify its importance.

I do not like the paper structure with the separate theoretical part "supplementary information." The theoretical part is good, but I would prefer if it can be combined with the first experimental part. Besides, to my opinion in the theoretical supplementary material the authors should demonstrate the gauge invariance of the Hamiltonian to prove the transition to BEC.

8) Finally, the English of the manuscript is not perfect. There are a set of minor typos and grammatical/stylistic flaws. In this light, it is highly recommended that the revised paper will be proofread by a English native-speaker.

Summary

While all the comments above are anticipated to be addressed in the authorial response and/or the revised manuscript, all of them do not deteriorate my - generally very positive - impression about this work, which I recommend for publication in Nature communications.

References

[1] S.A.Moskalenko, M.A.Liberman, E.S. Moskalenko, E.V.Dumanov, I.V. Podlesny, Coherence of Two-dimensional Electron-Hole Systems: Spontaneous breaking of continuous symmetry, A Review. Physics of the Solid State 55, 1563-11595 (2013).

[2] S.A. Moskalenko, M.A.Liberman, D.W.Snoke, E.V.Dumanov, S.S.Rusu, F.Cerbu, True, quasi and unstable Nambu-Goldstone modes of the two-dimensional Bose-Einstein condensed magnetoexcitons, Solid State Communications, 155, 57-61 (2013).

[3] S.A. Moskalenko, M.A. Liberman, E.S. Moskalenko, E.V. Dumanov, I.V. Podlesny, Coherent Two-dimensional Electron-Hole Systems: Spontaneous symmetry breaking, Solid State Physics (Fizika Tverdogo Tela -Russian). 55, 1457-1487 (2013).

[4] S.A. Moskalenko, M.A. Liberman, E.V. Dumanov, E.S. Moskalenko, Spontaneous symmetry breaking and coherence in two-dimensional electron-hole and exciton systems, Journal of Nanoelectronics and Optoelectronics, 7, 640 (2012).

[5] S.A. Moskalenko, M.A. Liberman, D.W. Snoke, E.V. Dumanov, S.S. Rusu, and F. Cerbu, Nambu-Goldstone modes of the two-dimensional Bose-Einstein condensed magnetoexcitons, Eur. Phys. J. B 85: 359 (2012).

[6] M.A. Liberman and A.V. Korolev, Bose condensation and superfluidity of excitons in semiconductors in high magnetic field, Phys. Rev. Letters, 72, 270 - 273 (1994).

[7] S.A. Moskalenko, M.A. Liberman, D.W. Snoke, V.V. Botsan and B. Johansson, Bose-Einstein condensation of excitons in ideal 2D system in a strong magnetic field, Physica E, Low dimensional Systems and Nanostructures, 19, No.3, 278-288 (2003).

Michael Liberman

We are grateful to the Reviewers for the thorough reading of the manuscript, the high estimation of our results, and the valuable remarks. Below are the answers to the Referees' criticism and comments.

Reviewer #1:

While the topic of this manuscript is very interesting, there are several points which have to be clarified. I suggest for the authors to demonstrate broader familiarity with the literature devoted to this topic.

1. For example, the following papers on dipolar magnetoexcitonic condensates in coupled quantum wells have not been cited:

- a. D. Yoshioka and A. H. MacDonald, J. Phys. Soc. Jpn. 59, 4211 (1990).
- b. Yu. E. Lozovik, O. L. Berman, and V. G. Tsvetus, Phys. Rev. B 59, 5627 (1999).

We thank the Referee for the useful comment and add the aforementioned references to the part of the paper where the bilayer condensate is mentioned.

2. The theory of condensation of dipolar magnetoexcitons in a graphene bilayers was reported in

O. L. Berman, Yu. E. Lozovik and G. Gumbs, Phys. Rev. B 77, 155433 (2008).

We thank the Referee for the useful comment and add the aforementioned papers to the bibliography (list of references).

3. The theory of condensation of microcavity magnetopolaritons formed by magnetoexcitons in either a semiconductor quantum well or a graphene monolayer, coupled to microcavity photons was presented in

O. L. Berman, R. Ya. Kezerashvili, and Yu. E. Lozovik, Phys. Rev. B 80, 115302 (2009).

We thank the Referee for the useful comment and add the aforementioned reference to the part of the paper where the polariton condensate is mentioned.

The authors should address how their results on condensation of magnetoexcitons are related to and different from the results, reported in the papers mentioned above.

We add a paragraph at the end of the manuscript clarifying the similarity and the difference between the observed fermionic condensate and the magnetoexciton condensate predicted in the aforementioned papers.

Reviewer #2:

While the authors use a model of noninteracting magnetoexcitons, I note that interactions between driven magnetoexcitons and interband excitons have been observed and studied before in the literature in the nonlinear four-wave-mixing signal during femtosecond timescales. Such time-resolved coherent spectroscopy experiments and the picture of nonlinear interactions between the magnetoexciton and exciton excitations of interest here might be useful to clarify the nature of this new state, since they are observed down to temperatures of Kelvin in the quantum Hall insulator.

We thank the Referee for the interesting observation. The four-wave-mixing experiments should in fact clarify the nature of the observed state. Unfortunately, our experimental setup does not allow realizing an experiment on femtosecond timescales right now due to the optical fibers used for photoexcitation of the electron system and collecting optical signals. However, we intend to perform a four-wave mixing experiment in future when a fiberless experimental setup is constructed.

Reviewer #3:

In this light, the present paper has a solid potential to be interesting to other researchers in the field and for a broad audience of the physics community. For this reason, I recommend it for publication, however, I do have some comments/suggestions that are necessary to making the manuscript more accessible and to improve the presentation, as summarized below.

1) While the authors pretended to provide a comprehensive overview in the Introduction section, I do not think that the current overview is satisfactory and the authors are unfair in their treatment of previous literature. In terms of the Bose-Einstein condensation of two-dimensional magnetoexcitons a bunch of novel experimental and theoretical papers appeared within the last decade is missed. There have been a comprehensive theoretical studies of the Bose-Einstein condensation (BEC) of two-dimensional magnetoexcitons, which in particular discussed the Berezinsky-Kosterlitz-Thouless phase transition at finite temperatures, and BEC of 2D magnetoexcitons for the conditions of GaAS, bilayer exciton condensates and heterostructures, published by Moskalenko, Liberman et al. [1-5]. The authors use the same name "magnetoexciton" for the photoexcited electron/vacancy complex therefore they must clearly explain the difference between their approach and the theory of BEC of 2D magnetoexcitons developed in [1-5]. References [1-5] are necessary, otherwise the text as it is presented is confusing for readers.

[1] S.A.Moskalenko, M.A.Liberman, E.S. Moskalenko, E.V.Dumanov, I.V. Podlesny, Coherence of Two-dimensional Electron-Hole Systems: Spontaneous breaking of continuous symmetry, A Review. *Physics of the Solid State* 55, 1563-11595 (2013).

[2] S.A. Moskalenko, M.A.Liberman, D.W.Snoke, E.V.Dumanov, S.S.Rusu, F.Cerbu, True, quasi and unstable Nambu-Goldstone modes of the two-dimensional Bose-Einstein condensed magnetoexcitons, *Solid State Communications*, 155, 57-61 (2013).

[3] S.A.Moskalenko, M.A.Liberman, E.S. Moskalenko, E.V.Dumanov, I.V. Podlesny, Coherent Two-dimensional Electron-Hole Systems: Spontaneous symmetry breaking, *Solid State Physics (Fizika Tverdogo Tela -Russian)*. 55, 1457-1487 (2013).

[4] S.A.Moskalenko, M.A.Liberman, E.V.Dumanov, E.S.Moskalenko, Spontaneous symmetry breaking and coherence in two-dimensional electron-hole and exciton systems, *Journal of Nanoelectronics and Optoelectronics*, 7, 640 (2012).

[5] S.A. Moskalenko, M.A. Liberman, D.W. Snoke, E.V. Dumanov, S.S. Rusu, and F. Cerbu, Nambu-Goldstone modes of the two-dimensional Bose-Einstein condensed magnetoexcitons, *Eur. Phys. J. B* 85: 359 (2012).

We thank the Referee for the critical remark. References [1-5] have avoided our attention. Now, we add them to the manuscript bibliography.

2) The first part of Introduction repeats (in fact it copies) what is written in the first half of the Abstract. I would suggest to remove lines 10-15 in the Abstract. In line 23 should be "...system of two-dimensional (2D) fermions"..

We have changed the abstract in accordance with the Referee's suggestion.

3) There are references [19], [23], [24] in the list of references, but there is no these references in the text. Ref. [19] is probably missed on lines 62/63 and so on.

This mistake occurred because of two independent bibliographies in our manuscript: one is intended for the main body of the manuscript, the other is for the Supplement. We have corrected this mistake in the new version of the manuscript.

4) Ref. [13] is probably missed on line 46, as well references on the publication by Moskalenko et al. are needed on line 44 concerning "Excitations in quantum Hall insulators are magnetoexcitons..."

We have added this reference.

5) All the same I believe that there is unnecessary citation [14] - (lines 51-53): "There are two types of magnetoexcitons: a spin singlet, $S = 0$, and a spin triplet, $S = 1$, with spin projections along the magnetic field axis $S_z = -1, 0, 1$ ", which is a simple statement, which does not require reference [14].

We have removed the unnecessary reference from the text.

6) Check Ref.[8] - line 157, should be [9] which is missed and [6, 7] in the list of references below.

As mentioned in the response to comment 3), this mistake was due to two independent bibliographies in our manuscript. We correct this mistake in the new version of the manuscript.

7) Line 134 the authors wrote: "This observation suggests that the theory developed in the previous paragraph is a reasonable approximation for the properties of the non-equilibrium system in question". I would suggest to change ".. the theory developed in the previous paragraph.. "for "..the estimate above..."

We accept the Referee's suggestion.

8) I do not like the paper structure with the separate theoretical part "supplementary information." The theoretical part is good, but I would prefer if it can be combined with the first experimental part.

We thank the Referee for the high estimation of the theoretical part presented in the Supplementary Information. We believe that the paper will become more difficult for some readers if the theoretical part is included into the main text. Since the Nature Communications is a journal for a broad scientific community we wish to retain the most complex part of the theory as a Supplementary Information for scientists who could easily follow the theoretical discussion.

9) Finally, the English of the manuscript is not perfect. There are a set of minor typos and grammatical/stylistic flaws. In this light, it is highly recommended that the revised paper will be proofread by a English native-speaker.

We have corrected the typos as suggested by the Referee. Before sending it to the Editor the manuscript had passed through correction made by the company affiliated with the Nature Publishing Group.